# Mass Spectrometry-Based Proteomics for Discovering Salivary Biomarkers in Periodontitis: A Systematic Review

**DOI:** 10.3390/ijms241914599

**Published:** 2023-09-27

**Authors:** Hongying Hu, Wai Keung Leung

**Affiliations:** 1State Key Laboratory of Oral Diseases, National Clinical Research Center for Oral Diseases, Department of Oral Medical Imaging, West China Hospital of Stomatology, Sichuan University, Chengdu 610041, China; summerhu0321@163.com; 2Faculty of Dentistry, The University of Hong Kong, Hong Kong SAR, China

**Keywords:** biomarkers, mass spectrometry, periodontitis, proteomics, review, saliva

## Abstract

Periodontitis is one of the primary causes of tooth loss, and is also related to various systemic diseases. Early detection of this condition is crucial when it comes to preventing further oral damage and the associated health complications. This study offers a systematic review of the literature published up to April 2023, and aims to clearly explain the role of proteomics in identifying salivary biomarkers for periodontitis. Comprehensive searches were conducted on PubMed and Web of Science to shortlist pertinent studies. The inclusion criterion was those that reported on mass spectrometry-driven proteomic analyses of saliva samples from periodontitis cohorts, while those on gingivitis or other oral diseases were excluded. An assessment for risk of bias was carried out using the Newcastle–Ottawa Scale and Quality Assessment of Diagnostic Accuracy Studies or the NIH quality assessment tool, and a meta-analysis was performed for replicable candidate biomarkers, i.e., consistently reported candidate biomarkers (in specific saliva samples, and periodontitis subgroups, reported in ≥2 independent cohorts/reports) were identified. A Gene Ontology enrichment analysis was conducted using the Database for Annotation, Visualization, and Integrated Discovery bioinformatics resources, which consistently expressed candidate biomarkers, to explore the predominant pathway wherein salivary biomarkers consistently manifested. Of the 15 studies included, 13 were case–control studies targeting diagnostic biomarkers for periodontitis participants (periodontally healthy/diseased, *n* = 342/432), while two focused on biomarkers responsive to periodontal treatment (*n* = 26 participants). The case–control studies were considered to have a low risk of bias, while the periodontitis treatment studies were deemed fair. Summary estimate and confidence/credible interval, etc. determination for the identified putative salivary biomarkers could not be ascertained due to the low number of studies in each case. The results from the included case–control studies identified nine consistently expressed candidate biomarkers (from nine studies with 230/297 periodontally healthy/diseased participants): (i) those that were upregulated: alpha-amylase, serum albumin, complement C3, neutrophil defensin, profilin-1, and S100-P; and (ii) those that were downregulated: carbonic anhydrase 6, immunoglobulin J chain, and lactoferrin. All putative biomarkers exhibited consistent regulation patterns. The implications of the current putative marker proteins identified were reviewed, with a focus on their potential roles in periodontitis diagnosis and pathogenesis, and as putative therapeutic targets. Although in its early stages, mass spectrometry-based salivary periodontal disease biomarker proteomics detection appeared promising. More mass spectrometry-based proteomics studies, with or without the aid of already available clinical biochemical approaches, are warranted to aid the discovery, identification, and validation of periodontal health/disease indicator molecule(s). Protocol registration number: CRD42023447722; supported by RD-02-202410 and GRF17119917.

## 1. Introduction

Periodontitis is a biofilm-associated, host-mediated inflammatory disease characterized by the progressive destruction of periodontal tissues [1]. It also ranks among the top chronic disorders globally. In 2010, severe periodontitis emerged as the sixth most prevalent health complication, with an impact on 10.8% of the global population, which equates to an estimated 743 million individuals [2]. According to findings from the National Health and Nutrition Examination Surveys (NHANES) between 2009 and 2014, the incidence of periodontitis in the United States was estimated to be 42.2% for adults aged between 30 to 79 years, where 7.8% of the cases suffered from its severe form [3]. Data from the Global Burden of Disease study indicated that the age-standardized prevalence rate of severe periodontitis was approximately 13% in 2019 [4]. Untreated periodontitis is a prominent factor contributing to tooth loss and edentulism, thereby adversely impacting quality of life [5,6]. Early diagnosis of and hence intervention in periodontitis is highly significant, as the loss of periodontal apparatus is progressive and largely irreversible [7].

Periodontitis diagnosis generally relies on information gathered from clinical and radiographic examinations. During a comprehensive clinical evaluation, the level of biofilm deposits, bleeding on probing (BOP), probing pocket depth (PPD), and probing attachment level are measured at six sites around every tooth [8]. Dental radiographs are also routinely employed to evaluate the extent of bone loss [9]. An accurate and precise diagnosis—both at a specific site and in an overall patient context—necessitates the documentation of various parameters, which are time-consuming and heavily depend on the examiner’s expertise. Furthermore, while these clinical parameters are presently the most effective measures for diagnosis, such data mainly reflects the history of periodontal destruction, while remaining ineffective at aiding clinicians to assess ongoing or predict future disease progression. Predictions about future disease activity remain the biggest challenge due to the low sensitivity and limited positive predictive value of these assessments [10].

In recent times, a new system for periodontitis classification and case definition, based on staging and grading, was proposed by the 2017 World Workshop [11]. Staging involves four categories (Stages I–IV), which are largely dependent on the severity of the disease, implying the complexity of the disease management. Grading, conversely, includes three levels (Grade A—low risk, Grade B—moderate risk, and Grade C—high risk for progression). At present, it only integrates factors that are associated with the severity of past and existing periodontal destruction, i.e., the most ‘reliable’ oral indicators for periodontitis progression, as well as the key general health status of diabetes metabolic control level and other exposures such as smoking [12]. It would be highly desirable to integrate thoroughly tested and established biomarkers into the case definition system to enhance diagnostic accuracy in the early detection of periodontitis while timely assessing the risk of future disease progression [12].

In previous studies that have investigated potential biomarkers for periodontitis, historical attention has predominantly focused on analysing gingival crevicular fluid (GCF), an inflammatory exudate emanating from the affected periodontal pocket. The GCF is known to encompass myriad potential biomarkers that are reflective of periodontal inflammation, including but not limited to prostaglandin E2, matrix metalloproteinase 8 (MMP8), interleukin-1beta (IL-1β), and interleukin-6 (IL-6) [13]. While commercial assays for some of these biomarkers are accessible [14], their incorporation into clinical practice remains scarce due to uncertainties about their predictive value, as well as added cost and time [15].

As an alternative, saliva provides a more efficient avenue for diagnostic endeavours, providing a less technically challenging and patient-friendly methodology compared to GCF [16]. For example, given the limited quantity of GCF obtained, typically less than 1 μL [17], measuring the concentration of candidate biomarkers presents a considerable challenge. Additionally, periodontal pathogens and host antibacterial proteins are easily detectable in saliva, thus rendering salivary-based diagnostics a promising avenue for periodontal diagnosis, chairside or even at a patient’s own home [18,19]. Therefore, utilizing salivary biomarkers for periodontitis screening, diagnosis, and predicting disease progression has been the subject of intensive research [16,20,21]. Yet, the usefulness of the respective biological molecules as an indicator for periodontitis onset/progression, as well as the grading biomarker for the new periodontitis classification system, remains inconclusive [12]. This ambiguity calls for an untargeted or unbiased approach, where multiple or possibly even hundreds of biological molecules are studied simultaneously with the aid of advanced bioinformatics and statistical techniques to facilitate data analysis.

In recent decades, the advent of ‘omics’ techniques, such as transcriptomics, proteomics, and metabolomics, has significantly advanced the salivary diagnostics field [22]. Among these, proteomics has claimed a pivotal position, which is largely attributable to the salivary milieu’s richness in proteins [23]. In contrast to conventional clinical biochemistry techniques such as enzyme assays or immunoassays, proteomics presents an edge in high-throughput analysis, which is an extremely important feature when unveiling novel biomarkers [24]. Employing mass spectrometry (MS)-based proteomics, more than 3000 unique proteins and peptides have been identified in saliva [25]. Intriguingly, the salivary proteome shares an estimated 30% of its proteins with both the plasma proteome and the GCF proteome [26,27], suggesting substantial potential for salivary diagnostics. Indeed, employing expression proteomics—a technique that focuses on a quantitative comparison of protein expression typically between pathologic and physiologic states, has, to date, resulted in the discernment of promising biomarkers demonstrating robust diagnostic value across various diseases [28,29,30,31], including periodontitis [32,33,34]. This emphasizes the potential of salivary proteomics for early disease detection, monitoring disease progression, and assessing treatment effectiveness. Thus, the present systematic review endeavours to render a holistic elucidation of the current application of MS-based salivary proteomics in the context of biomarker discovery for periodontitis. The replicability and expression of potential biomarkers for periodontitis were evaluated in terms of study cohorts and the type of diagnostic medium employed.

## 2. Materials and Methods

### 2.1. Protocol and Registration

The reporting of this review followed part of the Preferred Reporting Items for Systematic Review and Meta-analysis checklist, and the protocol was registered at the International Prospective Register of Systematic Review under the number CRD42023447722.

### 2.2. Eligibility Criteria

The inclusion criteria for relevant discovery-based articles were (1) studies that are either cross-sectional or longitudinal in nature and have been published in peer-reviewed scientific journals, (2) investigations centred around a periodontitis-affected cohort, (3) studies that utilized a whole saliva sample, and (4) studies employing mass-spectrometry as the primary tool for proteomic analyses.

The following articles were excluded: (1) reviews, letters, book chapters, conference abstracts, posters, or patents that are devoid of first-hand data; (2) studies that did not investigate differential protein/peptide expression; and (3) studies on periodontitis with other oral or systemic diseases/conditions.

### 2.3. Literature Search

A literature search was conducted on the PubMed and Web of Science databases to identify all existing studies on the application of salivary proteomics in periodontitis. The search was carried out without any restrictions on the publication date, but was limited to studies published in English. The keywords used for the search included combinations of terms such as “saliva” or “salivary”, “proteomics” or “proteome” or “mass spectrometry”, and “periodontitis” (the search strategy is summarised in Appendix A). Reviews, conference papers, and research studies were eligible for the initial screening process. The titles of the identified studies were first manually screened, followed by a review of the abstracts to determine the studies for full-text reading. Additionally, references and cited papers (Web of Science database) from included studies were examined to identify further relevant articles.

### 2.4. Study Selection

Initially, a screening process was conducted on the titles and abstracts of all included articles. Subsequently, the complete texts of studies which appeared to be relevant were scrutinized for eligibility. H.H. and a research assistant carried out the screening process independently and excluded publications that were deemed irrelevant. In case of any discrepancies, W.K.L. personally perused each paper with particular reference to the objectives of the current study, and decided whether to include or exclude it.

### 2.5. Risk-of-Bias Assessment

For case–control studies, individual study bias risk was meticulously evaluated, according to the Newcastle–Ottawa Scale (NOS), by H.H. and W.K.L. independently. Within the parameters of the NOS, studies could be allocated up to 9 stars, based on criteria including representativeness of subjects, comparability, and risk ascertainment. A study amassing ≥ 3 stars is typically categorized as being of low bias risk, and vice versa [35]. To further address potential oversights intrinsic to the NOS evaluation, a refined version of the Quality Assessment of Diagnostic Accuracy Studies (QUADAS-2) instrument [36] was integrated into this review. This ensured a rigorous assessment of bias risk, particularly concerning the diagnostic efficacy of biomarkers pertinent to periodontitis detection. For studies that veered from the conventional case–control framework, their quality was evaluated using the National Institutes of Health (NIH) quality assessment tool for before–after (pre-post) studies, i.e., case series reports, with no control group.

### 2.6. Data Analysis

Upon finalizing the studies for inclusion, data extraction was independently conducted by H.H. and a research assistant. The general information of the selected papers included authors’ names and year of publication. Information on participants including number of subjects, subgroups if any, age, gender, and ethnic or regional origin were reviewed. Moreover, the type of samples and the proteomic platform employed were retrieved. A thorough perusal was conducted on the differential expression findings reported from each investigation, including proteomic data both from the main text and Appendix A.

Identifying consistent candidate biomarkers across diverse cohorts suggests their potential significance, especially if the expression patterns are uniform. As such, the replicability of these salivary biomarkers was evaluated according to the number of cohorts that were reported. A replicable candidate biomarker was defined as a protein or peptide that was consistently identified from the same type of salivary sample, and differentially regulated in at least two independent cohorts, while maintaining the same trend (either upregulated or downregulated) in relation to periodontitis, when compared to healthy controls. The replicable candidate biomarkers were grouped by sample type and disease subgroups. The Gene Ontology enrichment analysis [37] conducted on these consistently expressed candidate biomarkers to explore the predominant pathway wherein salivary biomarkers consistently manifested. The analysis was performed using the Database for Annotation, Visualization, and Integrated Discovery (DAVID) bioinformatics resources [38].

## 3. Results

### 3.1. Results of Literature Search

After removing duplications and assessing the titles and abstracts of 187 potentially suitable publications, 50 full-text articles were screened for eligibility, and finally, 15 publications were eventually included in the present review for data extraction (Figure 1). These studies were published between 2009 and 2023 (Table 1). In summary, 13 studies used a case–control approach to identify salivary biomarkers for the diagnosis of periodontitis [27,32,33,34,39,40,41,42,43,44,45,46,47], while 2 studies investigated changes in the salivary proteome of periodontitis patients following non-surgical periodontal treatment [48,49].

### 3.2. Risk-of-Bias Assessment

The Newcastle–Ottawa Scale (NOS) was used to assess the quality of all shortlisted case–control studies (*n* = 13) [27,32,33,34,39,40,41,42,43,44,45,46,47]. Details of the NOS results are displayed in Appendix A. All the case–control studies (*n* = 13) were found to have a low risk of bias (score ≥ 3 stars) [35]. Six studies demonstrated a low risk of bias in terms of the representativeness of the cases, as they selected participants either consecutively or randomly [32,33,41,44,45,47]. Five studies showed high comparability by matching subjects based on age, gender, and the absence of systemic diseases [27,33,34,40,45], while eight studies showed moderate comparability by pairing control subjects solely based on the absence of systemic diseases (Appendix A). No study adopted secure record-keeping or a blinded interview (questionnaire) for ascertaining exposure factors, such as systemic diseases and smoking.

The results of the QUADAS-2 assessment are summarized in Appendix A. In terms of patient selection, all studies were deemed to have a high risk of bias, given that only participants with confirmed diagnoses were enrolled [36]. Regarding the index test domain, only one study by Bostanci et al. [42] showed a low risk of bias, as the proteomic analysis was conducted in a blinded manner. As for the reference standard (patient classification and assessment), as well as the flow and timing, all studies demonstrated a low risk of bias.

The NIH quality assessment tool for before-after (pre-post) studies with no control group was used to assess the quality of studies focusing on salivary protein profile changes after periodontal treatment (*n* = 2) [48,49]. Both studies were rated fair (Appendix A), with a total score of 7. These studies successfully presented clear objectives, inclusion criteria, interventions, and outcomes, and provided statistical analyses of results along with corresponding *p*-values. However, certain issues potentially introducing bias were identified. These included issues pertaining to subject recruitment, sample size calculation, blinding, and the absence of multiple outcome measures (Appendix A).

### 3.3. Description of Included Studies

Table 1 exhibits the included studies on the application of MS-based proteomic technologies for discovering biomarkers for periodontitis in saliva. Information on the cohort (sample size, gender, age, country/ethnicity), sample collected, proteomic platform, number of differentially expressed proteins, and potential biomarkers highlighted in the respective studies was documented.

#### 3.3.1. Study Population

The included case–control studies were from Asia: China, South Korea, (2 each); West Asia/South Europe: Türkiye (*n* = 1); Europe: France (*n* = 2), German (*n* = 1), Sweden (*n* = 1), United Kingdom (*n* = 1); and South America: Brazil (*n* = 3) (Table 1). Within the scope of the case–control studies centred on the diagnostic potential of salivary biomarkers, most studies utilized the criteria oulined by the 1999 International World Workshop for the Classification of Periodontal Disease and Conditions [50] for defining periodontitis. Case definitions from the 2005 5th European guidelines [51] and the 2017 World Workshop Classification [1] were adopted by only one [45] and two studies [27,47], respectively. Comprehensive details on the specific case definition criteria can be found in Appendix A.

The case–control studies predominantly focused on two clinical presentations: aggressive periodontitis (AP) and chronic periodontitis (CP), using the classification framework from 1999. Given the age distribution in the three subsequent studies—where subjects exhibited a mean age ranging from 43.8 to 64.2 years and employed the latter two classification criteria –these cohorts were subsequently designated under the ‘CP’ category for ensuing data analyses. Consequently, the included case–control studies consisted of five cohorts related to AP, comprising a range of 5 to 12 patients, and 11 cohorts associated with CP, encompassing a range of 5 to 107 patients.

The case series studies were from New Zealand and China (Table 1).

#### 3.3.2. Study Protocol

Variation in type of saliva sample utilized in proteomic analyses may lead to a distinct proteomic profile, consequently resulting in the identification of different biomarkers [26,52]. In the context of this review, unstimulated whole saliva (UWS) samples were collected for proteomic analysis in all five AP cohorts [32,34,40,41,42]. Conversely, the seven cohorts that focused on CP employed UWS samples [39,40,41,42,44,45,47], while the remaining four cohorts utilized stimulated whole saliva (SWS) samples [27,33,43,46]. Notably, in the two studies investigating the monitoring of periodontal treatment response, one study used SWS samples [48], while the other used UWS samples [49].

In terms of proteomic platforms, liquid chromatography tandem mass spectrometry (LC-MS/MS) and surface-enhanced laser desorption/ionization tandem time of flight mass spectrometry (MALDI-TOF/TOF) demonstrate exceptional sensitivity, precision, and high-throughput capabilities, making them invaluable tools in identifying salivary biomarkers associated with periodontitis [53]. In this review, LC-MS/MS was the most frequently used untargeted proteomic technique for biomarker discovery, being utilized in eight studies [32,33,34,42,45,46,48,49], while MALDI-TOF/TOF was used in four studies [39,40,44,47]. Beyond label-free proteomics, the isobaric tag for the relative and absolute quantification (iTRAQ) labelling proteomics method was incorporated in one study [27]. Additionally, targeted mass spectrometry strategies, such as selected reaction monitoring (SRM) and multiple reaction monitoring (MRM), were each implemented for biomarker discovery in one study [41,43]. Meanwhile, SRM was also utilized as a validation technique for potential biomarkers identified from LC-MS/MS in another investigation [42].

### 3.4. Biomarker Discovery

In this review, most studies reported the results as the number of differentially expressed peaks or spots of proteins/peptides, and further identified the specific corresponding proteins. However, the study by Antezack et al. [47] only documented the number of peptide peaks, and the study by Grant et al. (2022) [27] reported the proteomic results as protein clusters. It was noted that the number of differentially expressed proteins/peptides between individuals with periodontitis and healthy controls showed a striking heterogeneity (ranging from 2 to 100), and varied across different cohorts, saliva sample types, and proteomic platforms (Table 1).

For biomarker discovery studies, the area under the receiver operating characteristics (ROC) curve (AUC) is a crucial measure that evaluates the performance of a biomarker, depicting its ability to distinguish between those with or without the disease [54]. This metric ranges from 0.5, which indicates a test no better than random chance, to 1.0, signifying an excellent diagnostic test [55]. In this review, ROC curve analyses were performed in three studies [27,42,44].

The study conducted by Bostanci et al. [42] identified the top three proteins associated with periodontal disease (including AP, CP, and gingivitis) as matrix metalloproteinase 9 (MMP 9), Ras-related protein Rap-1A (RAP1A), and actin-related protein 2/3 complex subunit 5 (ARPC5). Conversely, the top proteins associated with periodontal health comprised clusterin (CLUS) and deleted in malignant brain tumour 1 (DMBT1). Notably, the protein pair of ARPC5 and CLUS showed the highest predictive efficacy, with an AUC of 0.97. Additionally, the protein pairs of RAP1A and DMBT1, and MMP 9 and DMBT1, demonstrated AUCs of 0.95 and 0.94, respectively, indicating high diagnostic accuracy.

Tang and co-workers [44] found that among a total of 91 salivary peptide peaks, 7 exhibited significant differences when comparing CP patients to healthy controls. Furthermore, the cumulative ROC curve analysis representing all differentially expressed peptide peaks in saliva yielded an AUC of 0.897.

Grant and colleagues [27] revealed that a combination of MMP9, alpha-1-acid glycoprotein (A1AGP), and pyruvate kinase (PK) demonstrated notable diagnostic accuracy with an AUC of 0.954 in differentiating CP patients from healthy individuals or those with gingivitis. Moreover, incorporating S100A8 as an additional candidate biomarker resulted in an enhanced diagnostic precision, leading to an increased AUC of 0.96.

### 3.5. Replicable Candidate Biomarkers

For effective disease screening and diagnosis, a biomarker must be reproducible, thereby enabling its broad applicability across populations and ultimate utility in a clinical setting [56]. Therefore, an evaluation of the reproducibility of candidate biomarkers was conducted across varied cohorts by scrutinizing the complete list of differentially expressed proteins (Appendix A) across diverse cohorts.

#### 3.5.1. Case–Control Studies

The final sample size of the included 13 studies were, at the most, modest (median *n* = 40).

Within the AP cohorts, five candidate biomarkers were observed consistently in two separate cohorts when UWS was used as the medium of diagnosis. Of these biomarkers, alpha-amylase (α-amylase) and serum albumin levels were found to be elevated in patients afflicted with AP, whilst carbonic anhydrase 6, immunoglobulin J chain, and lactoferrin were found to be diminished in comparison to the healthy control group.

When considering the CP cohorts wherein UWS was employed as the diagnostic medium, increased levels of alpha-amylase (α-amylase) and serum albumin levels in CP patients were reported across two different cohorts. In contrast, a consistent significant negative fold change was observed in carbonic anhydrase 6 levels in CP patients from two independent studies. On the other hand, with SWS as the diagnostic medium, four candidate biomarkers, specifically complement C, neutrophil defensin 1, profilin-1, and S100-P, consistently exhibited an elevation in levels in CP patients across two cohorts. Meanwhile, MMP9 in SWS was detectable at increased levels in two studies (1.5- or 4.3-fold) and at a lower level in another study [46], hence it was not followed further.

An attempt was made to conduct a Gene Ontology enrichment analysis for the replicable candidate biomarkers. However, the analysis did not yield any significant enriched terms for up-regulated or down-regulated salivary proteins. These replicable candidate biomarkers, along with their respective fold changes as reported in the corresponding studies, are listed in Table 2.

#### 3.5.2. Pre-Post Treatment or Case Series Studies

Studies investigating protein profile changes after non-surgical periodontal treatment varied in their choice of saliva sample (UWS or SWS), which may limit the comparability of the results. Despite these circumstances, a common protein, haptoglobin, was found in these studies to have a similar pattern of expression; it was significantly increased in active periodontitis compared to post-treatment. Notably, identifying reliable salivary biomarkers to monitor the response to non-surgical periodontal treatment will greatly enhance our ability to tailor and adjust therapeutic approaches based on individual patient response. However, the discovery and validation of such biomarkers remain largely unexplored, and warrant further investigation. Therefore, it is vital for future studies to consider the type of saliva sample used and strive for standardization to ensure the comparability of results.

## 4. Discussion

This study systematically reviewed the application of MS-based proteomics in the identification of salivary biomarkers for periodontitis. Expression proteomic techniques demonstrate substantial potential in uncovering salivary markers for diagnosing periodontitis, as evidenced by the significant differentiation of over 100 proteins between those with periodontitis and healthy controls (Table 1). Notably, this review highlights several candidate biomarkers that were consistently replicated across independent cohorts, namely salivary α-amylase, serum albumin, carbonic anhydrase 6, Immunoglobulin J chain, lactoferrin, component C3, neutrophil defensin, Profilin-1, and S100-P.

Salivary α-amylase is an enzyme that is essential in the carbohydrate digestion process [57]. In addition to its digestive function, the potential of α-amylase in saliva as a biomarker for various conditions has been explored, primarily due to its role as a stress response indicator [58]. For instance, salivary α-amylase levels were significantly higher in youths with anxiety disorders compared to healthy controls [59]. In the context of periodontitis, it has been revealed that patients with the disease exhibit significantly higher levels of salivary α-amylase activity in UWS when compared to healthy controls [60,61]. Additionally, a positive correlation has been established between salivary α-amylase activity and the number of teeth affected by periodontal disease with a probing pocket depth of 5 mm or greater [62]. The current review has identified a persistent increase in the expression of α-amylase in UWS in both AP and CP cohorts, which suggests a potential diagnostic value for this protein in periodontitis. However, the direct connection between salivary α-amylase and periodontitis remains controversial. One perspective posits that the inflammatory process intrinsic to periodontitis triggers an upsurge in α-amylase production within the salivary glands [63]. Alternatively, it has been hypothesized that the elevated levels of salivary α-amylase in periodontitis may be indicative of heightened physiological stress associated with chronic inflammation [60]. Notably, after periodontal treatment, a reduction in salivary alpha-amylase levels was observed, which was also accompanied by an improvement in clinical parameters [64].

Human serum albumin, the most abundant protein in plasma, plays an important role in regulating the colloidal osmotic pressure of blood [65]. Salivary albumin is considered a filtrate of serum in the oral cavity. Elevated levels of salivary albumin have been observed in patients medically compromised by, for example, immunosuppression, radiotherapy, diabetes, and oral cancers [66]. Moreover, increased levels of salivary albumin levels have been reported in patients with periodontitis [67,68,69]. Specifically, Henskens et al. [67] reported a positive correlation between salivary albumin levels and the concentration of total salivary proteins in patients with gingivitis and periodontitis. This correlation suggests that the observed rise in serum albumin in UWS might be attributed to plasma leakage due to inflammation. In this review, consistently elevated levels of UWS albumin were documented in both AP and CP cohorts. This underscores its potential viability as an imperative biomarker for assessing the severity and progression of periodontitis. Nevertheless, it is important to emphasize the modest sample sizes (spanning 5–10 subjects) of studies reporting this finding [32,39,40], necessitating cautious extrapolation to broader cohorts or populations.

Carbonic anhydrase 6, a zinc-containing metalloenzyme predominantly produced and secreted by salivary glands, is one of the major protein constituents of human saliva [70]. It catalyses the reversible hydration of carbon dioxide in saliva with a possible contribution to the pH homeostasis and taste perception in the oral cavity [71,72]. *CA IX* gene expression is upregulated by hypoxia through the HIF-1 activation cascade, and downregulated by the wild-type von Hippel–Lindau tumour suppressor protein [73]. Although consistently downregulated in UWS in cohorts of both AP and CP patients, as observed in this review, the existing literature does not directly establish a link between this protein and the pathogenesis of periodontitis. Instead, lower concentrations of salivary carbonic anhydrase 6 have been associated with dental caries [74,75] and primary Sjögren’s syndrome [76], suggesting a potential divergence of its role in various oral health conditions. The authors of this study suspect that perhaps failure/impaired salivary hypoxia responses, as reflected by downstream carbonic anhydrase 6 production, might predispose individuals to adult periodontitis.

The immunoglobulin J chain, a polypeptide that is essential for the structure and function of secreted IgA and IgM, has been implicated in facilitating the polymerization of these immunoglobulins and aiding their transport across mucosal surfaces, thus playing a significant role in mucosal immunity [77,78]. This review consistently observed the downregulation of the immunoglobulin J chain in UWS from AP patients. Nevertheless, using the UWS immunoglobulin J chain as a specific biomarker for periodontitis presents certain challenges. This is mainly due to the limited understanding of the association between the immunoglobulin J chain and periodontitis. Although certain studies have reported varying levels of salivary IgA, which is associated with the J chain, in periodontitis [79,80], these findings are yet to be consistently replicated in different investigations [81]. Additionally, the salivary J chain was found to be increased in diabetes patients with periodontitis compared to the controls (periodontally healthy diabetes patients) [82]. Therefore, further investigation is necessary to determine the potential of the salivary J chain as a reliable biomarker for periodontitis.

Lactoferrin, another immune-related protein, demonstrates consistent downregulation in UWS from patients with AP, as shown in Table 2. This iron-binding protein plays a crucial role in the innate immune system, exhibiting well-known antimicrobial effects against various bacteria, fungi, and viruses [83]. In particular, salivary lactoferrin contributes to maintaining symbiosis between the host and microbiome by regulating the oral microbiota [84]. Pertinently, it was documented that lactoferrin inhibits the proteinase activity of *Porphyromonas gingivalis*, specifically the inhibition of gingipain [85]. Given the pathogenicity of gingipain, lactoferrin may play a crucial role in preventing *P. gingivalis*-associated periodontitis. Furthermore, through its iron-sequestering properties, lactoferrin modulates the physiological balance of production of reactive oxygen species synthesis and their elimination rates [84], a dynamic balance that is crucial in periodontitis pathogenesis [86]. A significantly increased lactoferrin level with a fold change of 1.8 in CP SWS, however, was observed in one of the included studies [33].

An intriguing finding is that decreased salivary levels of lactoferrin have also been linked to Alzheimer’s diseases [87,88]. One study proposed that suboptimal salivary lactoferrin levels pave the way for oral dysbiosis, which in turn could be a precursor to Alzheimer’s disease [89]. Notably, previous research has yielded various results regarding lactoferrin levels in patients with periodontitis. In line with the findings from Salazar et al. 2013 [33], increased levels of lactoferrin in SWS were observed in subjects with localized AP [90] and CP [91,92] via techniques such as enzyme-linked immunosorbent assay (ELISA) or electroimmunoassay. Further studies are warranted to elucidate the precise levels of salivary lactoferrin in the milieu of periodontitis.

The complement system, particularly complement C3, serves as a core node of innate immunity, which is responsible for initiating and regulating immune and inflammatory responses [93]. Within this context, it is unsurprising that gingival inflammation in patients with periodontitis correlates with elevated complement C3 activity, while successful periodontal treatment results in decreased C3 activation [94,95,96]. This review also found complement C3 to be consistently upregulated in SWS in patients with CP. Previous studies have proposed complement C3 as a potential therapeutic target for periodontitis [97,98]. A Phase IIa clinical trial demonstrated that a once-weekly intra-gingival injection of the complement C3 inhibitor AMY-101 significantly reduced periodontal inflammation, as measured by the modified gingival index and BOP, along with MMP8 and MMP9 levels [99]. In a recent study, Damgaard et al. [96] compared the levels of complement C3 in UWS in patients diagnosed with Grade B or Grade C periodontitis with those in healthy control subjects, using ELISA. Their findings indicated elevated salivary levels of total C3 and C3dg complement fragment in patients with either Grade B or Grade C periodontitis compared to the healthy controls. Additionally, C3c levels were increased in patients with Grade C periodontitis. However, no significant difference was observed between patients with Grade B and Grade C periodontitis.

The present review also found that neutrophil defensin levels were consistently elevated in SWS in patients with CP, compared to healthy controls. Human defensins are divided into α- and β-defensin subfamilies, of which four types of α-defensins are predominantly found in neutrophils, which are hence referred to as neutrophil defensins [100]. In the UWS of healthy subjects, the concentrations of human neutrophil defensin 1 and defensin 2 were found to be 8.6 ± 8.0 μg/mL and 5.6 ± 5.2 μg/mL, respectively [101]. Alterations in salivary neutrophil defensin levels have been associated with oral diseases such as squamous cell carcinoma, lichen planus, glossitis associated with iron deficiency, and mandibular osteomyelitis [102,103,104]. Regarding periodontitis, recent studies have found that the levels of neutrophil defensins in SWS increased progressively with rising numbers of pocketed teeth, which can be observed in both the adult population aged between 40–60 years [105] and the elderly population aged 65 years or more [106]. The consistent elevation of salivary neutrophil defensin in patients with CP suggests its potential utility as a biomarker for periodontitis diagnosis [107]. Further validation and longitudinal studies are necessary to explore the feasibility of using salivary neutrophil defensin levels for early detection and disease progression monitoring.

The findings that profilin-1 and S100-P are consistently elevated in SWS in patients with CP are intriguing and invite further investigation. However, there is only a limited amount of literature that explains their roles in periodontitis, highlighting an area of potential research, in particular differentially increased expression of profilin-1 within UWS/SWS samples from CP (Table 1 and Table 2) [27,33]. Profilin-1 is involved in the regulation of actin dynamics, a crucial part of the cellular cytoskeleton [108], which may implicate cellular responses in periodontal inflammation or tissue remodelling. S100-P, on the other hand, is a calcium-binding protein known to play a role in cell proliferation, survival, and differentiation [109], which may be involved in the pathogenesis of periodontal disease. Investigating these proteins may offer novel insights into periodontitis pathogenesis and potentially aid in the early diagnosis and monitoring of the disease. Further studies are essential to validate these findings and to establish the utility of these biomarkers in clinical diagnostics.

The choice between using stimulated or unstimulated whole saliva as a diagnostic medium is pivotal, and can significantly influence salivary biomarker discovery. UWS is believed to provide a more equilibrated condition, with less influence from salivary glands and collecting devices. However, certain conditions characterized by reduced salivary flow, such as Sjögren’s syndrome or post-radiation scenarios, may necessitate the use of stimulated saliva collection to obtain optimal saliva volumes for proteomic analysis [110]. Golatowski et al. [111], in their investigation, compared the proteome profile of saliva collected through passive drooling (unstimulated) and drooling stimulated with paraffin gum or a cotton swab. Their findings indicated that the specific proteins identified are different among the collection approaches [111]. This observation was further supported by another study, which reported that saliva stimulation led to a reduction in proteins involved with immune response and inflammation process [112]. This divergence could predominantly be attributed to the sample type, although other variables spanning age, ethnic background, and choice of proteomic platform might have contributed. A recent systematic review focusing on the diagnostic accuracy of salivary biomarker for periodontitis reported a predominance of UWS as a diagnostic medium, representing 63.2% of the samples, while SWS made up a mere 21% [113]. Given these observations, the authors of this review advocate a tilt towards UWS when targeting biomarker detection for periodontitis in future investigations.

Another important factor that could contribute to the inconsistencies in biomarker discovery is the variability in the proteomic platform utilized. In this review, remarkable disparities emerged both in the number of differentially expressed proteins and the specific potential biomarkers across the diverse proteomic approaches, as explicated in Table 1. The employment of untargeted methodologies, as anticipated, yielded a higher number of differentially expressed proteins compared to targeted approaches. Furthermore, it appeared that LC-MS/MS outperformed MALDI-TOF/TOF in terms of the number of differentially expressed proteins identified [114]. Although MALDI-TOF/TOF stands as one of the most powerful tools for proteomic analysis, it possesses the inherent characteristics of being a relatively “soft” ionization technique that is well suited to the resolution of proteins with high-molecular weight (>100 kDa) [115]. Nevertheless, human biological mixtures such as whole saliva encompass a spectrum of both high-molecular weight and low-molecular weight proteins [52]. It is plausible that the limited mass window range of MALDI-TOF/TOF might partially contribute to the relatively lower number of differentially expressed proteins detected through this approach.

Another pivotal source of inconsistency in biomarker discovery can be attributed to variations within the study populations. The proteome of saliva, as with any other biological sample, is influenced by inherent physiological temporal changes [116]. In adults, a discernible inverse correlation between the secretion rates of UWS and age has been reported, both in males and females [117,118]. However, varied results were documented regarding the relationship between age and either the concentration of specific or total salivary proteins in the whole saliva [116,119,120]. In a cross-sectional study involving 187 subjects aged between 22 to 79 years, salient influences of both age and smoking habits on the salivary proteomic profile were observed. These factors emerged as being significant, even amidst the scrutiny of other potential determinants, such as sex, body mass index, and education level [121]. In light of these observations, it is imperative to incorporate controls that are matched for age and smoking status when designing a case–control study to explore potential disease biomarkers in saliva. By doing so, researchers can minimize the impact of these confounding factors and ensure the reliability of their findings in identifying relevant biomarkers that are associated with the disease of interest.

## 5. Limitations of Included Studies and Future Perspectives

Given the complexity and heterogeneity of periodontitis, it is unlikely that a single biomarker could stand out as a particular ‘factor’ that is consistently associated with periodontitis or favourable treatment outcomes. At present, only three included papers, two focusing on saliva proteomic of health vs. periodontitis [27,47] and the remaining one looking into salivary proteomic changes before and after periodontal treatment [49], applied the 2018 Periodontal Diseases Classification to their cohorts. Among these, only one study took the initiative to stratify periodontitis into Stage I/II and Stage III/IV [27]. As mentioned earlier, their findings suggest that a panel of MMP9, A1AGP, and PK, coupled with MMP8 and age, yielded a modest diagnostic precision, boasting an AUC of 0.789, when distinguishing advanced periodontitis (Stage III/IV) from its milder counterpart (Stage I/II). At present, no research has exclusively focused on protein panels in saliva that manifest differential expression across the various grades of periodontitis, using a proteomics approach. It remains speculative to assert whether specific salivary biomarkers might correlate with particular Grades of periodontitis and the corresponding favourable healing response.

Notwithstanding the aforementioned limitations, it was advocated that a combination of biomarkers could perhaps offer a promising strategy to better reflect disease heterogeneity. Based on this review, this approach has proven effective in several studies, demonstrating superior performance compared to single biomarkers [27,42,44]. Thus, proteomic methodologies hold significant promise for discovering salivary biomarker panels for periodontitis diagnosis.

However, despite the promising findings of mass spectrometry (MS)-based proteomics in salivary biomarker discovery for periodontitis, there are several generic limitations to consider. First, most of the included studies have employed relatively small sample sizes (n < 100), which may affect the statistical power and, consequently, the reliability of the results. Additionally, there is considerable variability across the included studies concerning the cohort characteristics, types of saliva samples used, proteomic platforms utilized, and statistical methods applied. This high level of heterogeneity has led to only a limited number of candidate biomarkers being consistently replicated across different studies (Table 2). Furthermore, while numerous potential biomarkers have been reported, the vast majority have not undergone successful validation, particularly through external validation cohorts or replication studies. This raises concerns about the universal applicability and reliability of these biomarkers. Considering the feasible detection of prospective biomarkers such as amylase, as well as serum albumin, through conventional clinical biochemistry techniques or established commercial kits (both of which are economically efficient and expedient), it is imperative for the focus to transition from solely identifying new biomarkers to adopting a dual approach. This approach should encompass high-throughput biomarker identification coupled with meticulous validation using cost-effective methodologies, which will ultimately pave the way for their integration into salivary diagnostics for periodontitis [122].

As mentioned previously, adopting the new classification system for periodontitis necessitates a standardization of case definitions in future research. This will enhance the consistency of research in this field and could potentially contribute to discovering more reliable biomarkers. Notably, while all of the included studies have aimed to find salivary biomarkers that could distinguish patients with periodontitis from healthy controls, none have specifically focused on novel biomarkers that are capable of informing and/or further refining the grade, i.e., the risk of periodontitis progression. Integrating salivary biomarkers into the exiting periodontitis Grading system could represent a significant advancement, potentially aiding the yet-to-be-resolved situation of real-time prediction and monitoring of disease progression. Therefore, further research in this area is warranted.

## 6. Conclusions

This systematic review highlights the substantial promise of salivary proteomics in identifying biomarkers for periodontitis. Several proteins have emerged as consistent candidate biomarkers across multiple independent cohorts. These findings could be useful for future validation studies to establish reliable biomarkers for periodontitis. Despite the promising results, it is crucial to acknowledge the existence of methodological limitations in the current body of work, including small sample sizes, lack of external validation for most candidate biomarkers, and high data heterogeneity due to variability in study designs and proteomic platforms. However, with rigorous study designs and robust validation processes, salivary proteomics could usher in a new era for point-of-care diagnostics in periodontitis.

## Figures and Tables

**Figure 1 ijms-24-14599-f001:**
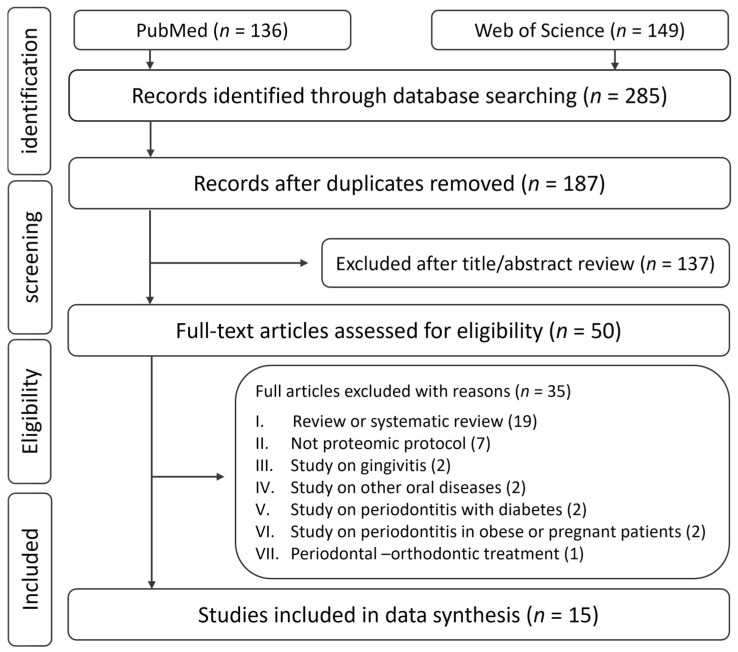
Flow diagram of the literature search and inclusion process. Details of search strategies are listed under Appendix A.

**Table 1 ijms-24-14599-t001:** Studies that applied mass spectrometry-based salivary proteomics for periodontitis biomarkers’ detection ^1^.

Author, Year	Cohort (*n*, F/M, Age Country/Ethnicity)	Sample	Proteomic Platform	DEP (*n*, Highlight) ^2^	Note
Diagnosis/Disease Association					
Wu et al., 2009 [32]	Healthy (*n* = 5, 2/3, 24.8 ± 3.83 years), Generalized AP (*n* = 5, 2/3, 24 ± 0.71 years) China	UWS	2DE-LC-MS/MS	*n* = 11 (protein spots) Increased (*n* = 7): lung and nasal epithelium carcinoma-associated protein 2, **serum albumin**, IgA2, zinc-α2 glycoprotein, IgC2, **α-amylase** (ranked 7th). Reduced (*n* = 4): elongation factor 2, **carbonic** **anhydrase 6**, 14-3-3σ, **lactoferrin**.	No result deposition in public domain or appendix to paper
Gonçalves et al., 2010 [39]	Healthy (*n* = 10, 5/5, 35.6 ± 9.5 years), CP (*n* = 10, 5/5, 45 ± 5.1 years). Brazil	UWS	SDS-PAGE, MALDI-TOF/TOF-MS	*n* = 4 Increased (*n* = 3): Ig heavy chain V-III region, **α-amylase**, **serum albumin**. Reduced (*n* = 1): cystatin-SN precursor.	No result deposition in public domain or appendix to paper
Kim et al., 2010 [40]	Healthy (*n* = 5, 3/2, 34.8 ± 2.9 years AP (*n* = 5, 3/2, 34.0 ± 4.0 years) CP (*n* = 5, 3/2, 34.6 ± 7.8 years) South Korea	UWS	2-DE-MALDI-TOF/TOF-MS	*n* = 4 (AP); 3 (CP) **AP vs. H**, increased: (*n* = 3): S100A9; **serum albumin**; lipocalin 1, reduced (*n* = 1): cystatin SN. **CP vs. H**, increased (*n* = 3): **serum albumin**; **α-amylase**, **profilin 1**, reduced: none.	No result deposition in public domain or appendix to paper
Salazar et al., 2013 [33]	Healthy (*n* = 20, 10/10, 48.6 ± 11.4 years), CP (*n* = 20, 10/10, 49.6 ± 10.2 years) Caucasian, German	SWS	LC-MS/MS	*n* = 20 Increased (*n* = 19): **S100-P**, plastin-2, **neutrophil defensin**, Rho GDP-dissociation inhibitor 2, catalase, **complement C3** (ranked 16th). Reduced (*n* = 1): lactoperoxidase.	Suppl. info.: doi/10.1111/jcpe.12130
Mertens et al., 2018 [41]	Healthy (*n* = 12, 8/4, 26.3 ± 4 years), AP (*n* = 11, 5/6, 33.3 ± 9 years), CP (*n* = 10, 4/6, 60.5 ± 9 years) France	UWS	LC-MRM	*n* = 2 (AP); 3 (CP) **AP vs. H**, increased (*n* = 2): hemopexin, fibrinogen α chain; reduced: none. **CP vs. H**, increased (*n* = 1): hemopexin; reduced (*n* = 2): polipoprotein H, plasminogen.	Suppl. info.: doi/suppl/10.4155/bio-2017-0218
Bostanci et al., 2018 [42]	Healthy (*n* = 16, unknown gender/age), AP (*n* = 17, ditto), CP (*n* = 17, ditto), Gingivitis (*n* = 17, ditto), Türkiye	UWS	LC-MS/MS, LC-SRM-MS	*n* = 100 (AP); 67 (CP) **AP vs. H**, increased (*n* = 37): RAS GTPase-activating-like protein, hemoglobin subunit alpha, glutaredoxin-1, S100-A4, hypoxanthine-guanine phosphoribosyl transferase; reduced (*n* = 63): Extracellular glycoprotein lacritin, isoform 1 of alpha-1-antichymotrypsin, calmodulin-like protein 5, isoform 1 of liver carboxylesterase 1, 1 family member 6, **carbonic anhydrase 6** (ranked 46th), **Immunoglobulin J chain** (ranked 60th). **CP vs. H**, increased (*n* = 5): Band 3 anion transport protein, ribonuclease R, ras gtpase-activating-like protein, proteasome activator complex subunit 2, metallo-beta-lactamase; reduced (*n* = 62): Isocitrate dehydrogenase cytoplasmic, isoform 1 of serpin b5, isoform 1 of histone deacetylase 5, calmodulin-like protein 5, isoform 1 of phospholipid transfer protein, **carbonic anhydrase 6** (ranked 34th).	Suppl. info.: doi/10.1074/mcp.RA118.000718
Grant et al., 2019 [43]	Healthy (non-smoker = 11, 6/5, 44.8 ± 12.3 years; smoker = 11, 5/6, 33.2 ± 11.2 years), Periodontal disease (non-smoker = 10, 5/5, 51.3 ± 17.8 years; smoker = 9, 4/5, 51.3 ± 15.0 years) Sweden	SWS	SRM-MS	*n* = 14 Increased (*n* = 8): **neutrophil defensin 1**, histone H2A type 2A, histone H2A type 2E, histone H2A type F-S, adrenomedullin. Reduced (*n* = 6): Ribonuclease 7, protachykinin 1, β-defensin 128, lipocalin 1, BPI fold-containing family B member 3.	Suppl. info.: doi/10.1159/000494146
Tang et al., 2019 [44]	Healthy (*n* = 16, 11/5, 33.1 ± 10.6 years), CP (*n* = 17, 9/8, 40.1 ± 10.9 years) China	UWS	MALDI-TOF/TOF-MS	*n* = 7 (peptide peaks) Increased (*n* = 2): Ig kappa variable 4-1, haptoglobin. Reduced: none.	Suppl. info.: doi.org/10.1016/j.cca.2019.04.076
Shin et al., 2019 [45]	Healthy (*n* = 100, 35/65, 64.2 ± 9.3 years), Periodontitis (*n* = 107, 36/71, 64.2 ± 9.0 years) South Korea	UWS	LC-MS/MS	*n* = 68 Increased (*n* = 33): neutrophil defensin 3, vitronectin, desmoplakin, vasodilator-stimulated phosphoprotein, Alpha-1B-glycoprotein. Reduced (*n* = 35): nucleobindin-2, poly polymerase 4, Ig kappa chain V-III region, hemopexin, 78 kDa glucose-regulated protein, **carbonic anhydrase 6** (ranked 24th)	Suppl. info.: doi.org/10.1007/s00784-018-2779-1
Hartenbach et al., 2020 [46]	Healthy (*n* = 10, 7/3, 29.9 ± 4.4 years, pooled 5 samples), CP (*n* = 30, 14/16, 42.0 ± 2.6 years, pooled 15 samples) Brazil	SWS	LC-MS/MS	*n* = 30 Increased (*n* = 3): cystatin-SA, salivary acidic PRP, submaxillary gland androgen-regulated protein 3B. Reduced (*n* = 27): keratin, type I cytoskeletal 13/4/2/9/16, cathepsin G, BPI fold-containing family B member 1, MMP9, annexin A1.	Suppl. info.: doi.org/10.1016/j.jprot.2019.103602
Antezack et al., 2020 [47]	Healthy (*n* = 74, 49/25, 24.50 ± 3.28 years), Periodontitis (*n* = 67, 53/14, 50.18 ± 13.85 years) France	UWS	MALDI-TOF/TOF-MS	*n* = 114 (peptide peaks) Only peptide peaks were analysed.	Suppl. info.: doi/10.1371/journal.pone.0230334
Grant et al., 2022 [27]	*Birmingham cohort:* Healthy (*n* = 10, 4/6, 39 ± 9 years), Stage I/II periodontitis (*n* = 10, 5/5, 47 ± 6 years), Stage III/IV periodontitis (*n* = 10, 6/4, 49 ± 7 years) *Newcastle cohort:* Healthy (*n* = 29, 16/13, 35 ± 11.9 years), Stage I/II periodontitis (*n* = 32, 15/17, 43.8 ± 7.2 years), Stage III/IV periodontitis (*n* = 28, 16/12, 43.8 ± 7.2 years) The United Kingdom	SWS	iTRAQ 8-plex labelling MS	*n* = 278 (protein clusters) Increased (*n* = 190): Haemoglobin subunit beta, Haemoglobin subunit alpha, Haemoglobin subunit delta, Haemoglobin subunit zata, AngRem52, **S100-P** (ranked 40th), **complement C3** (ranked 128th). Reduced (*n* = 88): Isoform V1 of Versican core protein, salivary proline-rich protein 2, 14-3-3σ isoform 2, actin-like protein, 14-3-3σ isoform 1.	Suppl. info.: doi.org/10.25500/edata.bham.00000684.
Casarin et al., 2023 [34]	Healthy (*n* = 13, 11/2, 37.0 ± 4.9 years), Generalized AP (*n* = 12, 10/2, 38.9 ± 14.4 years) Brazil	UWS	LC-MS/MS	*n* = 36 Increased (*n* = 21): Fibrinogen gamma chain, lactoperoxidase, **profilin-1**, heat shock protein beta-1, keratin, type I cytoskeletal 10, **α-amylase** (ranked 20th) Reduced (*n* = 15): Glutathione S-transferase P, keratin, type II cytoskeletal 4, leukocyte elastase inhibitor, alpha-2-macroglobulin, **immunoglobulin J chain**, **lactoferrin** (ranked 10th)	Suppl. info.: doi/10.1111/jcpe.13803
**Protein profile changes after treatment**					
Haigh et al., 2010 [48]	Generalized periodontitis before vs. after treatment (*n* = 9, 2/7, 35–66 years) New Zealand	SWS	SDS-PAGE, LC-MS/MS	*n* = 15 (protein spots) Increased (*n* = 8): transketolase, haptoglobin α-chain subunit, S100A8, S100-A9, S100-A6. Reduced (*n* = 2): parotid secretory protein, prolactin-inducible protein.	No result deposition in public domain or appendix to paper
Yuan et al., 2022 [49]	Stage I/II generalized periodontitis before vs. after treatment (*n* = 17, 8/9, 40.12 ± 11.60 years) China	UWS	LC-ESI-MS/MS	*n* = 9 (peptides) Increased (*n* = 3): Ig kappa variable 4-1, α-1-antitrypsin, haptoglobin. Reduced: none.	No result deposition in public domain or appendix to paper

2-DE: two-dimensional electrophoresis; AP: aggressive periodontitis; CP: chronic periodontitis; DEP: differentially expressed proteins/peptides. ESI: electrospray Ionization; F/M: female/male; H: healthy; iTRAQ: isobaric tag for relative and absolute quantitation; LC: liquid chromatography; MALDI: matrix-assisted laser desorption/ionization; MRM: multiple-reaction monitoring; MS: mass spectrometry; MS/MS: tandem MS; NA: data not available; PRP: proline-rich phosphoproteins; ROC: receiver operating characteristic; SDS-PAGE: sodium dodecyl sulphate–polyacrylamide gel electrophoresis; SELDI: surface-enhanced laser desorption/ionization; SRM: selected reaction monitoring; SWS: stimulated whole saliva; suppl. info.: supplementary information; TOF/TOF: tandem time of flight; UWS: unstimulated whole saliva. ^1^ Case and control definitions are shown in Appendix A. ^2^ DEP data arranged in descending order of absolute fold change reported in main text/supplementary materials. In instances where the number of DEPs exceeds five, only the top five, and/or the replicable candidate biomarkers (highlighted in bold, with rank), are enumerated. For an exhaustive list of DEPs, please refer to the Appendix A or the main text/Appendix A of the papers concerned.

**Table 2 ijms-24-14599-t002:** List of salivary biomarkers potentially associated with periodontitis ^1^.

Unstimulated Whole Saliva		Stimulated Whole Saliva		
Replicable Biomarkers	Expression (Fold Change ^2^)	Replicable Biomarkers	Expression (Fold Change ^2^)	References
*Aggressive periodontitis*				
Alpha-amylase	↑ (1.6, 1.4)			[32,34]
Serum albumin	↑ (1.5, 4.3)			[32,40]
Carbonic anhydrase 6	↓ (3.6, 2.3)			[32,42]
Immunoglobulin J chain	↓ (1.6, 1.8)			[34,42]
Lactoferrin	↓ (1.7, 1.3)			[32,34]
** *Chronic periodontitis* **				
Alpha-amylase	↑ (2.6, 6.1)			[39,40]
Serum albumin	↑ (2.17, 11.9)			[39,40]
Carbonic anhydrase 6	↓ (2.5, 1.4)			[42,45]
		Complement C3	↑ (1.7, 1.5)	[27,33]
		Neutrophil defensin	↑ (2.1, 3.3)	[33,43]
		Profilin-1	↑ (6.7, 1.6)	[27,33]
		S100-P	↑ (3.9, 2.4)	[27,33]

↑: increased expression; ↓: reduced expression. ^1^ Similar results reported in ≥2 independent cohort studies were included; ^2^ fold change of listed proteins in the corresponding reports.

## Data Availability

Not applicable.

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
