# Peer review of "Mass Spectrometry-Based Proteomics for Discovering Salivary Biomarkers in Periodontitis: A Systematic Review"

_ijms, 2023, doi:10.3390/ijms241914599_

Round 1
Reviewer 1 Report
It is known that periodontitis is one of the main causes of tooth loss and is associated with various systemic diseases. Early detection of this condition is critical to prevent further oral damage and associated health complications. Saliva, as a medium/carrier for proteins or biologically significant agents from various sources, offers a pool of potential putative specific biomarkers for the diagnosis and prognosis of numerous oral and systemic diseases/conditions. This review identified several putative biomarkers, including alpha-amylase, serum albumin, complement C3, MMP9, neutrophil defensin, profilin-1, and S100-P, all of which showed robust regulatory patterns in two independent cohorts/reports.
1. How reasonable is it to use proteomics methods to determine such indicators as amylase, albumin, etc., if there are well-proven methods of clinical biochemistry?
2. It seems to me that the authors should add a section in which to discuss the results of studies where patients with different stages of periodontitis were examined in order to understand which proteins increase their content with the progression of periodontitis.
Author Response
Reviewer 1
- How reasonable is it to use proteomics methods to determine such indicators as amylase, albumin, etc., if there are well-proven methods of clinical biochemistry?
Authors’ reply
We appreciate the comment and understand the concerns. Indeed, well-established clinical biochemistry methods exist for detecting some of these biomarkers, e.g., amylase and albumin. However, the rationale for employing proteomics methods in biomarker discovery is the nature of high-throughput analysis: proteomics allows for the simultaneous evaluation of hundreds and even thousands of proteins, enabling the identification of unique protein profile (panels) specific to periodontitis. This could be a more effective way to biomarker discovery than solely looking at individual proteins. Nevertheless, well-established clinical biochemistry techniques are relatively more cost-effective and easily accessible. While the proteomic approach is invaluable for research settings, especially when investigating new biomarkers or understanding complex biochemical interactions, it might be an overreach for routine diagnostics of established biomarkers. The comparative merits and demerits of proteomics versus traditional clinical biochemistry in this context were discussed (Page3, lines 111-114 and Page 12, lines 600-607).
Page3, lines 111-114
In recent decades, the advent of ‘omics’ techniques, such as transcriptomics, proteomics, and metabolomics, has significantly advanced the salivary diagnostics field [22]. Among these, proteomics has claimed a pivotal position, which is largely attributable to the salivary milieu’s richness in proteins [23]. In contrast to conventional clinical biochemistry techniques such as enzyme assays or immunoassays, proteomics presents an edge in high-throughput analysis, which is an extremely important feature when unveiling novel biomarkers [24]. Employing mass spectrometry (MS)-based proteomics, more than 3,000 unique proteins and peptides have been identified in saliva [25]. Intriguingly, the salivary proteome shares an estimated 30% of its proteins with both the plasma proteome and the GCF proteome [26,27], suggesting substantial potential for salivary diagnostics.
Page 12, lines 600-607
Considering the feasible detection of prospective biomarkers such as amylase, as well as serum albumin through conventional clinical biochemistry techniques or established commercial kits – both of which are economically efficient and expedient – it is imperative for the focus to transition from solely identifying new biomarkers to adopting a dual approach. This approach should encompass high-throughput biomarker identification coupled with meticulous validation using cost-effective methodologies, which will ultimately pave the way for their integration into salivary diagnostics for periodontitis [122].
- It seems to me that the authors should add a section in which to discuss the results of studies where patients with different stages of periodontitis were examined in order to understand which proteins increase their content with the progression of periodontitis.
Authors’ reply
Thank you for this insightful suggestion. We have added a section where findings from various studies examining patients across different stages of periodontitis were discussed (Page11, lines 570- 582).
Page11, lines 570- 582
At present, only three included papers, two focusing on saliva proteomic of health vs. periodontitis [27, 47] and the remaining one looking into salivary proteomic changes before and after periodontal treatment [49], applied the 2018 periodontal diseases classification on their cohorts. Among these, only one study took the initiative to stratified periodontitis into Stage I/II and Stage III/IV [27]. As mentioned earlier, their findings suggest that a panel of MMP9, A1AGP, and PK, coupled with MMP8 and age, yielded a modest diagnostic precision, boasting an AUC of 0.789, when distinguishing advanced periodontitis (Stage III/IV) from its milder counterpart (Stage I/II). At present, no research has exclusively focused on protein panels in saliva that manifest differential expression across the various Grades of periodontitis using proteomics approach. It remains speculative to assert whether specific salivary biomarkers might correlate with particular Grades of periodontitis and the corresponding favourable healing response.
Reviewer 2 Report
Dear authors, thank you for your manuscript. It's full of potential. However, it needs important changes.
You have chosen an important, but very broad field.
Abstract: authors report this sentence: "The advent of advanced "om-ics" technologies and bioinformatics methods when applied to salivary diagnostics could signifi- cantly enhance the utility of point-of-care screening or even home-based periodontitis detection."
The sentence is too vague and does not explain your objective. Many areas of research can be classified as omics. Examples include proteomics, transcriptomics, genomics, metabolomics, lipidomics and others. What is your focus?
You've chosen the narrative structure on a too wide subject. Risk of loss of value. You could provide a traditional review.
Introduction should be more focused.
Line 64-75: It's too focused on explaining biomarkers in general. We need to be more specific. Proteomics has three main types: expression proteomics, functional proteomics, and structural proteomics. Please, provide more information.
Table are well described.
The narrative structure penalizes the manuscript.
Minor editing of English language required
Author Response
- Abstract: authors report this sentence: "The advent of advanced "omics" technologies and bioinformatics methods when applied to salivary diagnostics could significantly enhance the utility of point-of-care screening or even home-based periodontitis detection."
The sentence is too vague and does not explain your objective. Many areas of research can be classified as omics. Examples include proteomics, transcriptomics, genomics, metabolomics, lipidomics and others. What is your focus?
Authors’ reply
Thanks very much for the comment. This sentence was deleted, and the aim of the present review as well as the inclusion criteria of included studies were described in the abstract (Page 1, lines 12-17).
Page 1, lines 12-17
This study offers a systematic review of the literature published up to April 2023, and aims to clearly explain the role of proteomics in identifying salivary biomarkers for periodontitis. Comprehensive searches were conducted on PubMed and Web of Science to shortlist pertinent studies. Inclusion criteria were those that reported on mass spectrometry-driven proteomic analyses of saliva samples from periodontitis cohorts, while those on gingivitis or other oral diseases were excluded.
- You've chosen the narrative structure on a too wide subject. Risk of loss of value. You could provide a traditional review.
Authors’ reply
We appreciate the comment. To address the concern of the narrative structure being too broad, we have restructured the manuscript to provide a traditional systematic review.
- Introduction should be more focused.
Authors’ reply
Thanks very much for the comment. We've made the introduction more focused on the core topic: the potential of proteomics in salivary diagnostics for periodontitis.
- Line 64-75: It's too focused on explaining biomarkers in general. We need to be more specific. Proteomics has three main types: expression proteomics, functional proteomics, and structural proteomics. Please, provide more information.
Authors’ reply
We appreciate the comment. The introduction of general biomarkers has been deleted to be more focused on biomarkers for periodontitis. We have elaborated the type of proteomics-expression proteomics-used in salivary biomarker discovery (Page 3, lines 118-123).
Page 3, lines 118-123
Indeed, employing expression proteomics – a technique that focuses on a quantitative comparison of protein expression typically between the pathologic and physiologic state, has, to date, resulted in the discernment of promising biomarkers demonstrating robust diagnostic values across various diseases [28-31], including periodontitis [32-34]. This emphasizes the potential of salivary proteomics for early disease detection, monitoring disease progression, and assessing treatment effectiveness.
- Table are well described.
Authors’ reply
Thanks for the comment.
- The narrative structure penalizes the manuscript.
Authors’ reply
We appreciate the comment. The structure has been revised into a standard systematic review.
- Minor editing of English language required.
Authors’ reply
The English has been edited by a technical writer.
Round 2
Reviewer 2 Report
No further comments